Whole-limb scaling of muscle mass and force-generating capacity in amniotes

Bishop Peter J. 1 2 pbishop@fas.harvard.edu
Wright Mark A. 1
Pierce Stephanie E. 1
1 Museum of Comparative Zoology and Department of Organismic and Evolutionary Biology,Harvard University , Cambridge, Massachusetts , United States of America
2 Geosciences Program, Queensland Museum , Brisbane, Queensland , Australia
Abdala Virginia
Electronic publication date: 2021 Nov 29
Publication date: 2021
Volume: 9
Electronic Location ID: e12574
Received 2021 Sep 2; Accepted 2021 Nov 9
Copyright: © 2021 Bishop et al.
Copyright year: 2021
Copyright holder: Bishop et al.
License: This is an open access article distributed under the terms of the Creative Commons Attribution License, which permits unrestricted use, distribution, reproduction and adaptation in any medium and for any purpose provided that it is properly attributed. For attribution, the original author(s), title, publication source (PeerJ) and either DOI or URL of the article must be cited.
License URL: https://creativecommons.org/licenses/by/4.0/

Keywords: Muscle architecture, Scaling, Allometry, Amniotes, Biomechanics

Funding: Harvard University United States National Science Foundation DEB-1754459 and EAR-2122115 Wetmore Colles Fund (Harvard University) This work was supported by the William F. Milton Fund (Harvard University, to Stephanie E. Pierce) and United States National Science Foundation grants DEB-1754459 and EAR-2122115 (to Stephanie E. Pierce), and is published with the assistance of a grant from the Wetmore Colles Fund (Harvard University, to Mark A. Wright). The funders had no role in study design, data collection and analysis, decision to publish, or preparation of the manuscript.

==============================
Skeletal muscle mass, architecture and force-generating capacity are well known to scale with body size in animals, both throughout ontogeny and across species. Investigations of limb muscle scaling in terrestrial amniotes typically focus on individual muscles within select clades, but here this question was examined at the level of the whole limb across amniotes generally. In particular, the present study explored how muscle mass, force-generating capacity (measured by physiological cross-sectional area) and internal architecture (fascicle length) scales in the fore- and hindlimbs of extant mammals, non-avian saurians (‘reptiles’) and bipeds (birds and humans). Sixty species spanning almost five orders of magnitude in body mass were investigated, comprising previously published architectural data and new data obtained via dissections of the opossum Didelphis virginiana and the tegu lizard Salvator merianae. Phylogenetic generalized least squares was used to determine allometric scaling slopes (exponents) and intercepts, to assess whether patterns previously reported for individual muscles or functional groups were retained at the level of the whole limb, and to test whether mammals, reptiles and bipeds followed different allometric trajectories. In general, patterns of scaling observed in individual muscles were also observed in the whole limb. Reptiles generally have proportionately lower muscle mass and force-generating capacity compared to mammals, especially at larger body size, and bipeds exhibit strong to extreme positive allometry in the distal hindlimb. Remarkably, when muscle mass was accounted for in analyses of muscle force-generating capacity, reptiles, mammals and bipeds almost ubiquitously followed a single common scaling pattern, implying that differences in whole-limb force-generating capacity are principally driven by differences in muscle mass, not internal architecture. In addition to providing a novel perspective on skeletal muscle allometry in animals, the new dataset assembled was used to generate pan-amniote statistical relationships that can be used to predict muscle mass or force-generating capacity in extinct amniotes, helping to inform future reconstructions of musculoskeletal function in the fossil record.

Introduction

Countless aspects of organismal biology vary with the size of the organism as a whole (Schmidt-Nielsen, 1985; Vogel, 2003). In a mechanical context, the differential scaling of lengths, areas, volumes, inertias and other metrics with increasing size (e.g., body mass) influences an organism’s ability to generate and resist forces. An oft-cited example is the comparison between the cross-sectional area of a tissue and the gravitational loads it must bear. Under isometric (geometric) scaling, area scales proportional to length2, or assuming a constant tissue density, mass2/3; in contrast, gravity-induced loads scale proportional to mass1. Thus, larger species would have disproportionately less tissue area to support their mass, incurring greater tissue stress (= force per unit area). Given that tissue mechanical properties are largely constant across species (e.g., yield strength of cortical bone; Currey, 2002), this scaling relationship is undesirable, and usually necessitates changes to organismal structure, behaviour or performance to mitigate the effects of increasing body size, such as larger species using a more extended limb posture during stance and gait (Biewener, 1989; Dick & Clemente, 2017).

The macroscopic architecture of skeletal muscle is one such aspect of organismal structure that is known to scale with body size in animals. A particularly relevant architectural metric is its physiological cross-sectional area (PCSA), which relates a muscle’s mass with the length and arrangement of its constituent fascicles (Calow & Alexander, 1973; Gans & Bock, 1965; Sacks & Roy, 1982), and can be used to estimate the muscle’s maximal force output in an isometric contraction (Powell et al., 1984; Medler, 2002). If animals were isometrically scaled, muscle mass would scale in direct proportion to total body mass, fascicle length would scale proportional to mass1/3, and PCSA—and therefore force-producing capacity—would scale proportional to mass2/3. Larger species would therefore be relatively weaker. Over the past five decades, numerous studies have examined the interspecific scaling of muscle mass and force-generating capacity (via PCSA) across a wide variety of species, in mammals (Alexander, 1977; Alexander et al., 1981; Bennett & Taylor, 1995; Cuff et al., 2016a, 2016b; Eng et al., 2008; McGowan, Skinner & Biewener, 2008; Pollock & Shadwick, 1994), birds (Bennett, 1996; Maloiy et al., 1979) and non-avian saurians (Allen et al., 2015; Cieri, Dick & Clemente, 2020; Dick & Clemente, 2016). Although the caveat must be acknowledged that different studies have used different line-fitting approaches to estimate scaling exponents, these studies have shown that across most species the architecture of individual muscles scales in a broadly similar fashion. Muscle mass typically exhibits weak negative to modest positive allometry (range of exponents 0.9–1.15), PCSA typically exhibits weak to modest positive allometry (range of exponents 0.69–0.91), and fascicle length typically exhibits modest positive to modest negative allometry (range of exponents 0.14–0.5). Some notable exceptions to this include the extreme positive allometry in the hindlimb extensors of macropod marsupials (mass exponents 1.11–1.61, PCSA exponents 0.78–1.33; Bennett & Taylor, 1995; McGowan, Skinner & Biewener, 2008) and many forelimb muscles in varanid lizards (mass exponents 0.95–1.48, PCSA exponents 0.75–1.15; Cieri, Dick & Clemente, 2020), and near mass independence of fascicle length in the plantaris of mammals (Pollock & Shadwick, 1994; although this may be affected by sampling in that particular study). The common occurrence of positive allometry helps reduce the ‘strength deficit’ experienced by larger animals, but this deficit is fully counteracted only in extreme cases of positive allometry, where the PCSA exponent exceeds 1.

It remains unknown as to how the aforementioned scaling patterns for individual muscles translate to scaling at the level of the whole limb. Likewise, it remains unknown as to whether architectural disparities observed between muscles, functional groups of muscles (e.g., ankle extensors) or even species remain at the level of the whole limb. The limbs of an animal must exist and function as a single integrated entity, which imposes at least two requirements on the constituent muscles: The muscles collectively share a common volume, whereby changes in the size of one muscle may affect the size of adjacent muscles (Fig. 1A). For example, in order for the limb to avoid becoming too heavy (or having too high a rotational inertia), increase in the size of one muscle may necessitate a decrease in the size of another.

Architectural specialization of muscles can limit their ability to effectively contribute to a diverse range of tasks (Wilson & Lichtwark, 2011), and hence specialization in one muscle may necessitate concomitant change in another so that limb functionality is not compromised. For example, a muscle with short fascicles will have a high PCSA but a reduced working range, which may require another muscle to compensate by having longer fascicles, but with reduced PCSA (Fig. 1B).

Figure 1 The requirements on muscles as part of a single, functionally integrated whole may impose constraints on their construction.

(A) Changes in the size of one muscle may necessitate change in the size of adjacent muscle, such that total muscle volume (illustrated here in cross-section) may remain relatively constant. (B) Functional specialization of one muscle’s architecture may necessitate concomitant changes to the architecture of other muscles, in order for a limb to remain capable of effectively executing a diverse range of tasks; total physiological cross-sectional area (PCSA) may therefore remain relatively constant. In this example, moving from left to right muscle 1 decreases PCSA but muscle 2 increases PCSA, such that total PCSA remains unaltered.

These two requirements imply that animal limbs are subject to a ‘fascicle packing problem’, and raise the question of how flexible (evolutionarily labile) animal limbs are in terms of their muscular anatomy. Are there different strategies for packing PCSA into a given volume of limb muscle, or do the differences observed between individual muscles and between species ‘cancel out’ at the level of the whole limb? Depending on how stringent the above requirements are, the total force-producing capacity (strength) of an animal’s limb may therefore be strongly tied to total limb muscle mass, irrespective of a given species’ size or functional requirements.

Understanding how whole-limb muscle mass and force-generating capacity relate to one another, and how this relation scales with body size, is not just pertinent to the study of extant species. Muscle force-generating capacity is a key unknown in studies of extinct animal function and behaviour, and empirical data from extant species play a vital role in informing inferences of extinct species (Bates & Falkingham, 2018; Bishop, Cuff & Hutchinson, 2021a; Fahn-Lai, Biewener & Pierce, 2020; Lautenschlager et al., 2018; Sellers et al., 2013). The use of empirical data from extant members of a specific clade may be appropriate for extinct members of that clade (e.g., using data derived from extant palaeognath birds to guide inferences of extinct palaeognath birds; Bishop, 2015), but it is not immediately clear how this should be approached for extinct species that are phylogenetically distant or morphologically disparate from extant species. In particular, the more phylogenetically distant or morphologically disparate an extinct taxon is, the lower the confidence that can generally be placed in inferences of muscle origin or insertion (Carrano & Hutchinson, 2002; Witmer, 1995), relative size, internal architecture and even whether a given muscle exists in the extinct taxon (e.g., differentiation from adjacent muscles). Developing an understanding of how muscle mass and force-generating capacity scales with body size across a wide range of species, even at a broad anatomical resolution such as the whole limb, could therefore provide a useful starting point for better informed inferences of extinct taxa. Even if just overall bulk and strength of limb musculature were able to be confidently constrained, this would represent a practical advance upon which future refinements could be made.

The present study sought to address the above outstanding issues by conducting a holistic assessment of whole-limb muscle mass and force-generating capacity in extant terrestrial amniotes. It had three key aims: (1) to contribute new data from hitherto unsampled clades (ameridelphian marsupials and lacertoid lizards), broadening the diversity of the available comparative dataset; (2) to investigate how muscles scale with body size at the level of the whole limb, in both the fore- and hindlimb, providing a first assessment of the stringency of the ‘fascicle packing problem’; (3) to derive generic, amniote-wide statistical predictive relationships that would have utility in deriving inferences of musculature in extinct amniote species. This study is the first comparative synthesis of muscle architecture scaling across extant mammals, birds and non-avian saurians. In addition to providing a novel perspective on the topic of muscle scaling in animals, the results from this study provides a platform for more rigorous inference of muscle strength in extinct amniote clades.

Materials & methods

Dataset

The present study is founded upon muscle architecture data derived from dissections (Table 1); all raw data used are provided in Table S1. The sources of the data are described in the following two subsections. Although this study addresses ‘whole limb’ scaling, it is restricted in scope to the musculature crossing the three primary joints of the limbs (shoulder/hip, elbow/knee and wrist/ankle), ignoring the intrinsic musculature of the manus and pes. This was necessitated by the practical difficulties of accurately dissecting and measuring the latter muscles (particularly for small species), such that data are very rarely reported in the comparative literature. Ignoring manual and pedal muscles is also justified given that they comprise a small fraction of total limb musculature, and presumably contribute only a small fraction towards limb support and propulsion during locomotor activities. Although extrinsic shoulder muscles attaching to the scapula can be important for locomotion in therian mammals (Hudson et al., 2011a; Jenkins & Weijs, 1979), these were excluded from consideration to facilitate a fair comparison across all species; only muscles that explicitly attached to the humerus or more distal forelimb skeleton were included. The forelimb dataset comprised architectural measurements for a total of 912 muscles in 31 species (21 mammals and 10 non-avian saurians, the latter hereafter referred to as ‘reptiles’ for simplicity) spanning more than four orders of magnitude in body mass, and the hindlimb dataset comprised measurements for a total of 1,181 muscles in 36 species (19 mammals, 12 reptiles and five birds) spanning almost five orders of magnitude in body mass. In the hindlimb dataset, birds and humans were collectively treated as a single group, ‘bipeds’, such that ‘mammals’ herein refer to all mammals except humans. Given the small sample size (and taxonomic skew) of birds in the current dataset, and the broad aims of the study, it was not considered justifiable to analyze birds and humans as separate entities here. Furthermore, as the present study concerns terrestrial amniotes, the forelimb of bipeds was not investigated.

Table 1 Summary of data sources used in the study.

Species	Group	Body mass (kg)	Data	Study	
Osteolaemus tetraspis	Reptile	10.2	Fore, hind	Allen et al. (2015)	
Crocodylus johnstoni	Reptile	20.19	Fore, hind	Allen et al. (2015)	
Crocodylus moreletii	Reptile	27.7	Fore, hind	Allen et al. (2015)	
Alligator mississippiensis	Reptile	57.7	Fore, hind	Allen et al. (2015)	
Melanosuchus niger	Reptile	90	Fore, hind	Allen et al. (2015)	
Crocodylus niloticus	Reptile	278	Fore, hind	Allen et al. (2015)	
Varanus tristis	Reptile	0.173	Hind	Dick & Clemente (2016)	
Varanus varius	Reptile	2.936	Hind	Dick & Clemente (2016)	
Varanus panoptes	Reptile	4.15/1.6	Fore, hind	Dick & Clemente (2016), Cieri, Dick & Clemente (2020)	
Varanus komodoensis	Reptile	40/37.3	Fore, hind	Dick & Clemente (2016)	
Varanus spenceri	Reptile	2.098	Fore	Cieri, Dick & Clemente (2020)	
Salvator merianae	Reptile	1.359	Fore, hind	This study	
Basiliscus vittatus	Reptile	0.0511	Hind	Bergmann & Hare-Drubka (2015)	
Eudromia elegans	Biped	0.545	Hind	Bishop et al. (2021b)	
Numida meleagris	Biped	1.45	Hind	Cox et al. (2019)	
Gallus gallus (junglefowl)	Biped	2.079	Hind	Paxton et al. (2010)	
Dromaius novaehollandiae	Biped	42	Hind	Lamas, Main & Hutchinson (2014)	
Struthio camelus	Biped	105	Hind	Smith et al. (2006)	
Martes martes	Mammal	1.55	Fore	Böhmer et al. (2018)	
Taxidea taxus	Mammal	7.6	Fore	Moore et al. (2012)	
Canis familiaris	Mammal	31.4/27	Fore, hind	Williams et al. (2008), Ellis, Rankin & Hutchinson (2018)	
Felis nigripes	Mammal	1.1	Fore, hind	Cuff et al. (2016a, 2016b)	
Felis silvestris	Mammal	2.66	Fore, hind	Cuff et al. (2016a, 2016b)	
Leopardus pardalis	Mammal	9.6	Hind	Cuff et al. (2016b)	
Caracal caracal	Mammal	6.6	Fore, hind	Cuff et al. (2016a, 2016b)	
Panthera uncia	Mammal	36	Fore, hind	Cuff et al. (2016a, 2016b)	
Panthera onca	Mammal	44	Fore, hind	Cuff et al. (2016a, 2016b)	
Panthera tigris	Mammal	86	Fore, hind	Cuff et al. (2016a, 2016b)	
Panthera leo	Mammal	133	Fore, hind	Cuff et al. (2016a, 2016b)	
Mus musculus	Mammal	0.02345	Hind	Charles et al. (2016)	
Rattus norvegicus	Mammal	0.323	Hind	Eng et al. (2008)	
Marmota monax	Mammal	4.7	Fore	Rupert et al. (2015)	
Equus caballus	Mammal	510	Hind	Payne et al. (2005)	
Ceratotherium simum	Mammal	2160	Fore, hind	Etienne, Houssaye & Hutchinson (2021)	
Rhinoceros unicornis	Mammal	2065	Fore, hind	Etienne, Houssaye & Hutchinson (2021)	
Lepus europeus	Mammal	3.454	Fore, hind	Williams, Payne & Wilson (2007), Williams, Wilson & Payne (2007)	
Scalopus aquaticus	Mammal	0.1062	Fore	Rose et al. (2013)	
Rangifer tarandus	Mammal	91	Fore, hind	Wareing et al. (2011)	
Homo sapiens	Biped	75.337	Hind	Rajagopal et al. (2016)	
Pan troglodytes	Mammal	54.7	Hind	Carlson (2006)	
Bradypus variegatus	Mammal	3.97	Fore	Olson et al. (2018)	
Dasypus novemcinctus	Mammal	3.6	Fore	Olson et al. (2016)	
Isoodon fusciventer	Mammal	1.329	Fore	Martin et al. (2019)	
Didelphis virginiana	Mammal	1.759	Fore, hind	This study	
Tachyglossus aculeatus	Mammal	3.79	Fore	Regnault et al. (2020)	
Note:

List of species with architectural data for individual muscles, the major group they belong to (for the purposes of the statistical analyses conducted here), body mass, whether a given species contributed to the forelimb (‘fore’) or hindlimb (‘hind’) datasets, and studies in which the data were originally published. Note that birds and humans were analysed together in this study as ‘bipeds’. See Table S1 for additional species and studies that contributed data on muscle mass only.

For each muscle, its physiological cross-sectional area (PCSA) was calculated as

(1) PCSA=mmuscle⋅cos(αO)ρ⋅ℓO,

where mmuscle is belly mass, αo is pennation angle, ℓo is fascicle (or ‘fibre’) length and ρ is muscle tissue density, the latter taken to be constant for vertebrate skeletal muscle at 1,060 kg/m3 (Hutchinson et al., 2015; Mendez & Keys, 1960). It is important to note that this equation assumes that all fascicles of a muscle act in parallel in generating force, allowing their individual cross-sectional areas to be summed (see also Sacks & Roy, 1982). In reality, the constituent fibres of a given fascicle are often shorter than the fascicle itself, wherein their ends overlap or interdigitate (Gaunt & Gans, 1992; Infantolino, Neuberger & Challis, 2012), although they may still activate simultaneously with one another, thus functioning as a single fibre (Bodine et al., 1982). Additionally, as per common practice, it was assumed that measurements of fascicle length and pennation angle corresponded to the muscle’s optimal fibre length and optimal pennation angle, respectively (Zajac, 1989).

Published data

The majority of data used in this study were sourced from previously published comparative studies on muscle architecture, biomechanics or scaling (see Table 1 for references; see also Martin et al., 2020 for a review of methods used to measure muscle architecture). Unfortunately, many of the earlier studies on muscle architecture scaling cited in the Introduction did not report their raw measurements and so their data were unable to be incorporated into a taxonomically richer analysis. Literature data were selected according to three requirements: All or almost all of the muscles of the limb had been measured and reported (save the manual and pedal muscles, as noted above), since the overarching aim of the present study is considering the whole limb. Studies in which a few small muscles (e.g., deep muscles such as the gemellus, quadratus femoris or popliteus) were not reported were still included, since the omission of such small muscles is expected to have minimal effect on the overall results. Studies that measured multiple muscles under a single name (e.g., multiple heads of the flexor carpi radialis as a single muscle) were also included, since this nevertheless accounts for all the muscle mass present. In contrast, studies that did not report one or more major muscles were excluded from consideration. All datasets ultimately selected for use in the present study included all major extensor and adductor (antigravity) muscles of the limb. The specific muscles included (and which, if any, were excluded) in a given source study are outlined in Table S1.

Each of the architectural parameters listed in Eq. (1) were either explicitly reported, or their values able to be back-calculated from reported values, the latter either as PCSA (e.g., Cuff et al., 2016a, 2016b; back-calculated via Eq. (1)) or mass-scaled values (e.g., Dick & Clemente, 2017; Cieri, Dick & Clemente, 2020; back-calculated by unscaling according to reported body mass). A few previous studies had neglected to measure (or at least report) pennation angle, but to maximize consistency across the present dataset these studies were excluded. Studies that reported dry muscle mass only were also excluded.

The data reported were for adult or large-sized individuals, to reduce potential confounding effects of ontogenetic variation (Table S1). In approximately two-fifths of species sampled, multiple similarly-sized individuals had been investigated, but the data reported by the relevant studies were only presented as a species mean, wherein a given architectural parameter for a given muscle had already been averaged across the individuals studied; in these cases, the mean body mass of that sample was used. For all other species, architectural and body mass data were reported for separate individuals. When data for multiple individuals of a given species had been reported separately (e.g., Lamas, Main & Hutchinson, 2014; Allen et al., 2015; Martin et al., 2019), those for the largest individual were used, to reduce possible ontogenetic effects. This approach was deemed more appropriate than computing a species average across all individuals, because in several instances the sample of individuals investigated by prior studies (especially those focused on ontogeny) exhibited high disparity in body sizes, undermining the value of a species mean; moreover, such an arithmetic mean would not account for ontogenetic allometry within the species, and hence could introduce further error into the analyses. Ultimately, each species contributed only a single datapoint to the analyses.

Data were also sourced for a further 15 species from six additional studies, which had reported just muscle mass (Table S1). These studies either reported mass of each individual muscle separately (Hudson et al., 2011a; Hudson et al., 2011b; Ogihara et al., 2009; Payne et al., 2006; Zihlman, McFarland & Underwood, 2011), or reported total muscle mass for the limb as a whole (Grand, 1977), and helped further increase taxonomic coverage in the final dataset.

New data

To broaden the taxonomic diversity of the dataset, and contribute novel data that future studies may draw upon, dissections were undertaken on a single adult individual each of the Virginia opossum (Didelphis viginiana) and the Argentine black and white tegu (Salvator merianae). The former is an ameridelphian marsupial, and apart from a single species of bandicoot (australidelphian; Martin et al., 2019) is the only other marsupial in the dataset; the latter is a lacertoid lizard, representing a hitherto unsampled part of squamate phylogeny. Intact whole carcasses of wild individuals were obtained as part of a prior study (Fahn-Lai, Biewener & Pierce, 2020), sourced from Worcester County, Massachusetts (D. virginiana, Massachusetts Division of Fisheries and Wildlife) and Everglades National Park, Florida (S. merianae, Daniel Beard Center, United States Geological Survey). Upon acquisition, the specimens were immediately frozen at −20 °C, and subsequently fully thawed prior to dissection and architectural measurement. The right fore- and hindlimbs were dissected in both cases. Muscle architecture measurement followed standard dissection procedures (Allen et al., 2015; Biewener & Full, 1992; Fahn-Lai, Biewener & Pierce, 2020), using a magnifying lamp where necessary. To minimize desiccation of the fresh specimens, tissues were kept moistened with paper towel soaked in saline solution throughout. Muscle belly mass was measured with an electronic balance (Precisa 320 XB; Precisa Gravimetrics AG, Dietikon; 0.0001 g precision), fascicle length with digital calipers (0.01 mm precision) and pennation angle with a transparent protractor (1° precision). Measurements of fascicle lengths and pennation angles were made at random locations throughout a muscle belly, and depending on muscle size up to ten measures each were made, after which the arithmetic mean was derived.

Anatomical comparisons

Four main, and two subsidiary, comparisons were undertaken for both fore- and hindlimbs, as described below. For each comparison, patterns were examined for the limb as a whole, as well as just the proximal and distal limb muscles separately, given that some previous studies have noted scaling differences between the proximal and distal limb for individual muscles or functional groups (Alexander et al., 1981; Cuff et al., 2016a, 2016b; Dick & Clemente, 2016; Eng et al., 2008). ‘Proximal’ muscles were classified as those in which the majority of their bulk resides proximal to the elbow (forelimb) or knee (hindlimb) joints, whereas ‘distal’ muscles have the majority of their bulk residing distal to those joints. This volume-based approach is more relevant to the ‘fascicle packing problem’, and avoids the complications caused by flexor or extensor muscles crossing the joints involved. Furthermore, given that a substantial component of hindlimb locomotor muscle in reptiles is the caudofemoralis longus (CFL), which chiefly resides in the proximal tail, whole hindlimb and proximal hindlimb analyses were also re-run with this muscle excluded from the reptile sample. In addition to providing more nuanced insight into questions of scaling and fascicle packing, these additional variations provide greater sophistication to predictive relationships derived from the data. Ultimately, eight different sub-analyses were performed for each anatomical comparison (48 in total).

#1—Total muscle mass (Σmmuscle) v. body mass (mbody)

This comparison examines how much of total biomass is invested in limb musculature; under isometry, the scaling exponent would be 1. It ignores the potential for systematic variation in relative body composition in terms of other tissue types (bone, integument, etc.), but is nonetheless informative because it focuses on one of the tissues primarily involved in body support and propulsion during terrestrial locomotion.

#2—Mean size-normalized isometric strength v. mbody

Following calculation of PCSA as per Eq. (1), the arithmetic mean across all muscles was taken. Although PCSA is an estimator of maximal isometric force-generating capacity of a muscle, as a measure of area it is not a particularly intuitive descriptor of force, in and of itself. To present mean PCSA in a more tangible form, it was converted to maximal isometric force in multiples of body weight (BW):

(2) Fmax∗=PCSA⋅σmbody⋅g,

where σ is maximal isometric stress, assumed here as 300,000 N/m2 (Bates & Falkingham, 2018; Hutchinson, 2004; Medler, 2002), and g is the acceleration due to gravity. Dividing by body mass means that under isometry the scaling exponent is that of PCSA minus 1.0 (i.e., −⅓), which in turn makes it more straightforward to interpret in the context of scaling. That is, PCSA can scale with positive allometry and yet force-generating capacity can decline with increasing size if ⅔ < PCSA exponent < 1, whereas the sign of the exponent for Fmax* is immediately indicative of whether a strength deficit exists at larger body size: only if it is positive is relative force-generating capacity (i.e., relative performance) maintained with increasing size. It should be noted that using a different value for σ would not alter the resulting scaling exponent.

#2b—Median Fmax* v. mbody

The relative masses and force-generating capacities of individual limb muscles typically do not follow an even or symmetrical distribution (Table S1), such that the analysis of means as above may be influenced by one or two exceedingly strong, or weak, muscles in the limb. Thus, the median PCSA for all muscles was also computed, and converted to Fmax* as per Eq. (2). Under isometry, the scaling exponent would also be –⅓.

#2c—Total Fmax* v. mbody

In the current dataset there is considerable variation in the number of individually measured and reported muscles for a given species. These differences can reflect investigator judgement in the splitting or grouping of muscle heads for measurement, but can also be due to legitimate anatomical differences between species; for example, crocodylians have a single long digital flexor in the hindlimb, whereas birds have up to six. Variation in the number of muscles may influence the mean or median PCSA, and so to account for this the total PCSA was also computed and converted to Fmax* as per Eq. (2). Again, under isometry the scaling exponent would be −⅓.

#3—Characteristic fascicle length v. mbody

Fascicle length is almost ubiquitously investigated in studies of muscle architecture scaling, yet in a broad comparison across amniotes it is not sensible to investigate fascicle length in and of itself, either for individual muscles, functional groups or for the whole limb. This is due to the great variation that can exist in muscle size, origins, insertions and lengths, as well as differences of subdivision (differentiation) between different taxonomic groups, which can all lead to marked variation in fascicle length irrespective of body size. An alternative approach is to compute a single ‘characteristic fascicle length’ for the limb as a whole, as the weighted harmonic mean of the fascicle lengths of each individual muscle (cf. Alexander et al., 1981):

(3) L*=Σmmuscleρ⋅ΣPCSA.

This effectively replaces the musculature of the whole limb (or proximal or distal compartment thereof) with a single equivalent muscle. Note that in using PCSA, Eq. (3) weights fascicle lengths by both muscle mass and cos(αo), and thus the equivalent whole-limb muscle also factors in pennation. Furthermore, it can be seen that L* is approximately inversely proportional to mean relative (mass-normalized) PCSA (Eng et al., 2008). As for ‘real’ fascicle length, the scaling exponent under isometry would be ⅓.

#4—ΣPCSA v. Σmmuscle

By removing the context of mbody, this provides a direct assessment of the ‘fascicle packing problem’, by testing how whole-limb force-generating capacity compares against the amount of available muscle mass. Under isometry, the scaling exponent would be ⅔; note that isometry may reflect scale invariance of pennation angle (which would be theoretically expected; see also Dick & Clemente, 2017; Cieri, Dick & Clemente, 2020), fascicle length, or a more complex combination of both parameters. Comparison #3 also addresses the fascicle packing problem, but indirectly.

Statistical analyses

All data processing and analyses were conducted in R v.4.1.0 (R Core Team, 2021), using the ‘caper’ (v.1.0.1; Orme et al., 2018), ‘evomap’ (v 0.0.9; Smaers & Rohlf, 2016) ‘phytools’ (v.0.7; Revell, 2012), ‘geiger’ (v.2.0.7; Harmon et al., 2008) and ‘nlme’ (v. 3.1; Pinheiro et al., 2021) packages; the full set of code and data used are provided in the Supplemental Information. Data were logarithmically transformed (base 10) prior to analysis, facilitating the use of linear statistical models. Phylogenetically informed statistical analyses were conducted using a single, fully resolved, time-calibrated phylogenetic tree of the study taxa (Fig. S1). The tree was generated using the TimeTree database (www.timetree.org; Hedges, Dudley & Kumar, 2006), which included all taxa except for Isoodon fusciventer; this was substituted with Isoodon obesulus (the only species of Isoodon in the database), which has no effect on divergence times with respect to other taxa in the present study.

For each of the above anatomical comparisons (for whole-limb and proximal and distal compartments), allometric scaling equations of the form log10Y = log10A + Blog10X were derived for mammals, reptiles and bipeds separately, using phylogenetic generalized least squares (pGLS or ‘phylogenetic regression’; Smaers & Rohlf, 2016) to determine slopes (exponents, B) and intercepts (log10A). This simultaneously estimated the λ parameter of Pagel (1999) using maximum likelihood, offering greater flexibility than a strict Brownian motion model of evolution in accounting for phylogenetic signal in the data. Additionally, the 95% confidence interval (CI) of the slope was determined, using the t-distribution, to facilitate comparison against the slope expected under isometry: if the expected exponent fell outside of the CI, the scaling was deemed to significantly depart from isometry at the P = 0.05 level. Subsequently, all groups were collated together and a new pGLS fit was computed to generate a ‘pan-amniote’ regression for a given anatomical comparison (again for whole-limb and proximal and distal compartments), thus deriving a statistical predictive framework that can be applied to extinct species. In addition to estimator coefficients, 95% CIs and prediction intervals were also derived (Smaers & Rohlf, 2016), which can provide error margins for future estimations.

To ascertain whether mammals, reptiles or bipeds exhibited different allometric trajectories for a given anatomical comparison, a phylogenetic analysis of covariance (pANCOVA) was performed, which tested for differences in slope and intercept between groups (both separately and together; Smaers & Rohlf, 2016). Here the off-diagonal elements of the variance–covariance matrix were scaled by the λ parameter estimated during calculation of the pan-amniote regression, to minimize false negatives caused by an overly conservative assumption of strict Brownian motion. The analyses were also run without accounting for phylogeny (ANCOVA, λ = 0), to evaluate the effect of phylogenetic relatedness on the results, given that the mammal, reptile and biped samples are nearly mutually exclusive phylogenetically (Fig. S1). Statistical significance was set at P = 0.05.

Results

Scaling results and exponents are presented graphically in Figs. 2–5; as the results for mean Fmax* (comparison #2) were highly similar to those for median and total Fmax* (comparisons #2b and #2c, respectively), the latter are presented in the Supplemental Information (SI text, Figs. S2, S3). The results for hindlimb analyses excluding the caudofemoralis longus (CFL) from the reptile (i.e., non-avian saurian) dataset are presented in Figs. S4–S9. The results of inter-group comparisons of allometries are presented in Tables 2–5, with those for median and total Fmax* presented in the Supplemental Information (Tables S2, S3); the results for hindlimb analyses excluding the CFL from the reptile dataset are presented in Tables S4–S9. Coefficients for the pan-amniote regressions are reported in Table 6, with coefficients for hindlimb analyses excluding the CFL from the reptile dataset presented in Table S10.

Figure 2 Total muscle mass v. body mass across extant terrestrial amniotes.

This is shown for the forelimb and hindlimb, in terms of the limb as a whole, and parsed by proximal and distal compartments. Results are plotted on logarithmic coordinates, along with phylogenetic regression (pGLS) and 95% confidence intervals (CIs). Red = mammals, blue = reptiles, green = bipeds. Note the difference in vertical and horizontal scales. The slope expected under isometry is indicated in each case by the solid grey line, plotted clear of the data for ease of viewing. The slope and 95% CIs for each group are reported in each case, as are the sample sizes (in parentheses); also indicated are instances where a particular group exhibits statistically significant positive (⇧) or negative (⇩) allometry.

Figure 3 Mean size-normalized isometric strength v. body mass across extant terrestrial amniotes.

Conventions as in Fig. 2. See Figs. S2 and S3 for the results of median and total size-normalized isometric strength v. body mass.

Figure 4 Characteristic fascicle length v. body mass across extant terrestrial amniotes.

Conventions as in Fig. 2.

Figure 5 Total physiological cross-sectional area v. total muscle mass across extant terrestrial amniotes.

Conventions as in Fig. 2.

Table 2 Results of comparisons between each major group via pANCOVA, for the analysis of Σmmuscle v. mbody.

Anatomical region	Test	Mammals v. reptiles	Mammals v. bipeds	Reptiles v. bipeds	
		df	F	P	F†	P†	df	F	P	F†	P†	df	F	P	F†	P†	
Forelimb	Whole	S	2.29	1.442	0.24	32.465	<0.001											
I	2.29	1.639	0.211	33.14	<0.001											
S + I	2.28	1.08	0.353	19.112	<0.001											
Proximal	S	2.30	1.008	0.324	25.295	<0.001											
I	2.30	1.114	0.3	27.243	<0.001											
S + I	2.29	0.73	0.491	14.967	<0.001											
Distal	S	2.31	0.481	0.493	24.355	<0.001											
I	2.31	0.509	0.481	17.047	<0.001											
S + I	2.30	0.377	0.689	11.945	<0.001											
Hindlimb	Whole	S	2.42	1.673	0.203	14.715	<0.001	2.36	5.181	0.029	11.763	0.002	2.15	21.811	<0.001	41.475	<0.001	
I	2.42	0.259	0.613	13.985	0.001	2.36	2.616	0.115	10.143	0.003	2.15	8.332	0.011	32.61	<0.001	
S + I	2.41	0.855	0.433	9.291	0.001	2.35	2.519	0.095	6.554	0.004	2.14	13.11	0.001	36.147	<0.001	
Proximal	S	2.32	4.268	0.047	16.724	<0.001	2.26	3.688	0.066	6.595	0.016	2.15	7.192	0.017	10.129	0.006	
I	2.32	0.787	0.382	9.695	0.004	2.26	2.362	0.136	5.666	0.025	2.15	3.064	0.101	8.641	0.01	
S + I	2.31	2.112	0.138	8.643	0.001	2.25	1.831	0.181	3.564	0.044	2.14	3.521	0.058	5.825	0.014	
Distal	S	2.32	0.089	0.768	8.089	0.008	2.26	9.577	0.005	26.166	<0.001	2.15	10.088	0.006	15.239	0.001	
I	2.32	0.199	0.659	15.57	<0.001	2.26	5.05	0.033	26.659	<0.001	2.15	6.737	0.02	31.079	<0.001	
S + I	2.31	0.166	0.848	7.819	0.002	2.25	4.64	0.019	17.517	<0.001	2.14	6.955	0.008	17.242	<0.001	
Note:

Each pairwise comparison was tested for differences in slope (S), intercept (I) and slope and intercept (S + I). Results for analyses without controlling for phylogeny are also presented (ANCOVA, †); significant results are in boldface; df = degrees of freedom.

Table 3 Results of comparisons between each major group via pANCOVA and ANCOVA, for the analysis of mean Fmax* v. mbody.

Anatomical region	Test	Mammals v. reptiles	Mammals v. bipeds	Reptiles v. bipeds	
		df	F	P	F†	P†	df	F	P	F†	P†	df	F	P	F†	P†	
Forelimb	Whole	S	2.25	1.212	0.282	9.013	0.006											
I	2.25	2.436	0.131	11.512	0.002											
S + I	2.24	1.187	0.322	5.725	0.009											
Proximal	S	2.26	0.673	0.419	6.247	0.019											
I	2.26	1.831	0.188	8.6	0.007											
S + I	2.25	0.881	0.427	4.183	0.027											
Distal	S	2.27	1.604	0.216	9.006	0.006											
I	2.27	1.554	0.223	8.939	0.006											
S + I	2.26	0.942	0.403	4.824	0.017											
Hindlimb	Whole	S	2.27	6.699	0.015	6.699	0.015	2.21	14.358	0.001	14.358	0.001	2.15	35.374	<0.001	35.374	<0.001	
I	2.27	0.886	0.355	0.886	0.355	2.21	6.785	0.017	6.785	0.017	2.15	6.387	0.023	6.387	0.023	
S + I	2.26	3.674	0.039	3.674	0.039	2.20	6.919	0.005	6.919	0.005	2.14	16.722	<0.001	16.722	<0.001	
Proximal	S	2.27	4.379	0.046	4.379	0.046	2.21	10.455	0.004	10.455	0.004	2.15	19.785	0.001	19.785	0.001	
I	2.27	0.238	0.629	0.238	0.629	2.21	5.882	0.024	5.882	0.024	2.15	4.273	0.056	4.273	0.056	
S + I	2.26	2.818	0.078	2.818	0.078	2.20	5.117	0.016	5.117	0.016	2.14	9.453	0.003	9.453	0.003	
Distal	S	2.27	2.232	0.147	12.439	0.002	2.21	11.022	0.003	14.949	0.001	2.15	49.529	<0.001	92.834	<0.001	
I	2.27	0.096	0.76	7.245	0.012	2.21	3.055	0.095	6.169	0.022	2.15	3.88	0.068	16.093	0.001	
S + I	2.26	1.075	0.356	6.432	0.005	2.20	5.402	0.013	7.131	0.005	2.14	23.132	<0.001	47.383	<0.001	
Note:

Conventions as per Table 2.

Table 4 Results of comparisons between each major group via pANCOVA and ANCOVA, for the analysis of L* v. mbody.

Anatomical region	Test	Mammals v. reptiles	Mammals v. bipeds	Reptiles v. bipeds	
		df	F	P	F†	P†	df	F	P	F†	P†	df	F	P	F†	P†	
Forelimb	Whole	S	2.25	1.537	0.227	1.537	0.227											
I	2.25	0.271	0.607	0.271	0.607											
S + I	2.24	1.151	0.333	1.151	0.333											
Proximal	S	2.26	1.251	0.274	1.251	0.274											
I	2.26	0.58	0.453	0.58	0.453											
S + I	2.25	0.642	0.535	0.642	0.535											
Distal	S	2.27	0.055	0.816	0.263	0.613											
I	2.27	0.113	0.74	0.288	0.596											
S + I	2.26	0.062	0.94	1.921	0.167											
Hindlimb	Whole	S	2.27	0.395	0.535	0.025	0.875	2.21	0.379	0.545	0.073	0.789	2.15	0.603	0.45	0.013	0.91	
I	2.27	0.181	0.674	0.157	0.695	2.21	0.651	0.429	1.437	0.244	2.15	0.114	0.741	0.243	0.629	
S + I	2.26	0.211	0.811	0.223	0.802	2.20	1.658	0.216	2.24	0.133	2.14	0.84	0.452	0.313	0.736	
Proximal	S	2.27	0.002	0.964	0.794	0.381	2.21	0.269	0.609	0.175	0.68	2.15	0.041	0.842	0.094	0.764	
I	2.27	0.008	0.929	0.367	0.55	2.21	0.273	0.607	0.372	0.548	2.15	0.346	0.565	0.636	0.438	
S + I	2.26	0.008	0.992	0.383	0.685	2.20	0.84	0.446	0.966	0.398	2.14	0.44	0.653	0.354	0.708	
Distal	S	2.27	3.644	0.067	2.953	0.097	2.21	0.64	0.433	3.941	0.06	2.15	1.323	0.268	0.14	0.713	
I	2.27	0.277	0.603	5.02	0.034	2.21	1.846	0.189	11.95	0.002	2.15	0.422	0.526	1.106	0.31	
S + I	2.26	1.782	0.188	2.519	0.1	2.20	0.928	0.412	5.75	0.011	2.14	1.663	0.225	0.635	0.545	
Note:

Conventions as per Table 2.

Table 5 Results of comparisons between each major group via pANCOVA and ANCOVA, for the analysis of ΣPCSA v. Σmmuscle.

Anatomical region	Test	Mammals v. reptiles	Mammals v. bipeds	Reptiles v. bipeds	
		df	F	P	F†	P†	df	F	P	F†	P†	df	F	P	F†	P†	
Forelimb	Whole	S	2.25	2.729	0.111	2.729	0.111											
I	2.25	2.081	0.162	2.081	0.162											
S + I	2.24	1.591	0.225	1.591	0.225											
Proximal	S	2.26	0.937	0.342	0.937	0.342											
I	2.26	0.901	0.351	0.901	0.351											
S + I	2.25	0.572	0.572	0.572	0.572											
Distal	S	2.27	0.01	0.92	3.383	0.077											
I	2.27	0.689	0.414	3.327	0.079											
S + I	2.26	0.434	0.653	1.726	0.198											
Hindlimb	Whole	S	2.27	0.075	0.787	0.075	0.787	2.21	5.272	0.032	5.272	0.032	2.15	1.215	0.288	1.215	0.288	
I	2.27	1.265	0.271	1.265	0.271	2.21	0.225	0.64	0.225	0.64	2.15	0.642	0.435	0.642	0.435	
S + I	2.26	0.611	0.551	0.611	0.551	2.20	2.676	0.093	2.676	0.093	2.14	1.124	0.353	1.124	0.353	
Proximal	S	2.27	0.004	0.951	0.004	0.951	2.21	2.84	0.107	2.84	0.107	2.15	0.655	0.431	0.655	0.431	
I	2.27	1 × 10−4	0.993	1 × 10−4	0.993	2.21	0.313	0.582	0.313	0.582	2.15	0.028	0.869	0.028	0.869	
S + I	2.26	0.002	0.998	0.002	0.998	2.20	1.743	0.201	1.743	0.201	2.14	0.392	0.683	0.392	0.683	
Distal	S	2.27	1.466	0.237	3.118	0.089	2.21	1.027	0.323	2.911	0.103	2.15	4.137	0.06	0.146	0.708	
I	2.27	0.505	0.483	9.851	0.004	2.21	0.195	0.663	3.387	0.08	2.15	0.233	0.636	0.738	0.404	
S + I	2.26	1.628	0.216	4.994	0.015	2.20	0.551	0.585	2.141	0.144	2.14	3.242	0.07	1.062	0.372	
Note:

Conventions as per Table 2.

Table 6 Pan-amniote regression (pGLS) coefficients for each comparison.

Comparison	Slope under isometry	Limb	Whole limb	Proximal	Distal	
Slope	Intercept	Mean %PE	Slope	Intercept	Mean %PE	Slope	Intercept	Mean %PE	
Σmmuscle v. mbody	1.0	Fore	1.0009	−1.5459	64.8	1.0076	−1.6512	66.52	0.9786	−2.3141	83.01	
Hind	0.9798	−1.2764	40.77	0.9975	−1.3619	32.84	0.9069	−2.0589	55.32	
Mean Fmax* v. mbody	−0.333	Fore	−0.2798	0.1213	43.11	−0.2871	0.2092	46.12	−0.276	−0.0573	45.03	
Hind	−0.3177	0.2791	33.58	−0.3266	0.3149	35.28	−0.3046	0.1345	45.66	
Median Fmax* v. mbody	−0.333	Fore	−0.3045	−0.0375	38.71	−0.3174	0.0782	44.67	−0.282	−0.1878	46.31	
Hind	−0.2916	0.0154	40.38	−0.2757	0.0449	44.48	−0.3027	−0.0581	42.02	
ΣFmax* v. mbody	−0.333	Fore	−0.2896	1.5756	48.65	−0.2755	1.4068	48.2	−0.3146	1.0427	66.34	
Hind	−0.307	1.7461	40.79	−0.2988	1.6095	36.11	−0.3314	1.1473	67.41	
L* v. mbody	0.333	Fore	0.2559	−1.6124	23.32	0.2603	−1.5407	21.54	0.2607	−1.8702	41.57	
Hind	0.2943	−1.5699	21.46	0.3016	−1.5117	21.91	0.2296	−1.7568	35.41	
ΣPCSA v. Σmmuscle	0.666	Fore	0.7344	−1.7894	18.82	0.735	−1.8876	18.46	0.7215	−1.7914	38.93	
Hind	0.7141	−1.8341	26.48	0.7095	−1.9253	26.2	0.746	−1.795	36.97	
Note:

These are reported for data on a log10 scale; also reported is the mean percent prediction error (%PE), expressed in original (antilog) terms. Results for hindlimb analyses where the caudofemoralis longus was excluded from the reptile dataset are reported in Table S10.

Total muscle mass v. body mass

Reptiles almost ubiquitously show negative allometry, whereas mammals and bipeds do not show significant departure from isometry (Fig. 2). The one exception to this generalization is negative allometry in the distal hindlimb of mammals: larger mammals tend to have relatively lighter (less muscled) distal hindlimbs. When the CFL is excluded, the values of the slopes (and CIs) for reptiles change minimally, but this is sufficient to render the revised scaling statistically indistinguishable from isometry (Fig. S4). Analyses of covariance (without accounting for phylogeny) indicate that mammals, reptiles and bipeds each exhibit significantly different allometric trajectories, in terms of both slope and intercept (Tables 2, Table S4). Reptiles have a lower slope and intercept compared to mammals, indicating that they have proportionately lower muscle mass, especially at large body sizes. In contrast, bipeds have proportionately greater hindlimb muscle mass than both mammals and reptiles, especially at larger body size and particularly in the distal limb. Many of these differences disappear (i.e., become statistically non-significant) once phylogeny is taken into consideration using the pANCOVA.

Mean size-normalized isometric strength ( Fmax*) v. body mass

Both mammals and reptiles generally do not show significant departure from isometry, although reptiles exhibit negative allometry in the distal hindlimb (Fig. 3). When the CFL is excluded, the values of the slopes for reptiles change minimally, but narrower CIs render the revised scaling significantly negatively allometric, for both the whole and proximal hindlimb (Fig. S5). Notably, bipeds exhibit strong positive allometry throughout the hindlimb, particularly in the distal limb (Fig. 3); notwithstanding the small sample size, mean size-normalized force-generating capacity appears to be scale-invariant. Analyses of covariance without accounting for phylogeny indicate that mammals, reptiles and bipeds frequently exhibit significantly different allometric trajectories, especially in terms of slope (Table 3, Table S5). Differences in intercept were mostly detected for the forelimb, where reptiles exhibit a markedly lower intercept than mammals, indicating lower mean force-generating capacity regardless of body size. Most of the differences between mammals and reptiles disappear once phylogeny is taken into consideration using pANCOVA; in contrast, most differences between mammals and bipeds, and reptiles and bipeds, were retained following phylogenetic correction, attesting to strong allometric deviation in the biped sample.

Characteristic fascicle length (L*) v. body mass

In the forelimb, both mammals and reptiles exhibit negative allometry, although in reptiles this is not statistically significant in the distal limb, due to wide CIs in that instance (Fig. 4). Mammals and bipeds also exhibit negative allometry in the hindlimb, although in mammals this is driven by the distal limb only; mirroring patterns noted above, bipeds show a stronger departure from isometric scaling in the distal limb. Reptiles do not show any significant departure from isometry in the hindlimb, a result that remains unaltered when the CFL is excluded from the dataset (Fig. S8). In stark contrast to the previous comparisons, ANCOVA (without accounting for phylogeny) revealed almost no significant difference among mammals, reptiles or bipeds (Table 4, Table S8). The only difference detected (which disappeared following phylogenetic correction using pANCOVA) was that mammals have a lower intercept in the distal hindlimb compared to both reptiles and bipeds, indicating that, overall, the distal hindlimb of mammals possesses proportionately shorter muscle fascicles.

Total PCSA v. total muscle mass

Mammals and bipeds almost ubiquitously exhibit positive allometry, whereas reptiles generally do not exhibit statistically significant departures from isometry (Fig. 5). Two exceptions to this are mammals not showing a significant departure from isometry in the distal forelimb, and reptiles exhibiting positive allometry in the forelimb as a whole. Excluding the CFL does not significantly alter the reptile hindlimb scaling patterns (Fig. S9). As for L*, ANCOVA (without accounting for phylogeny) notably revealed almost no significant difference among mammals, reptiles or bipeds (Tables 5, Table S9). Mammals have a lower slope in the hindlimb overall compared to bipeds, although this cannot be attributed to differences within the proximal or distal compartments. Additionally, mammals have a higher intercept in the distal hindlimb compared to reptiles (a result which disappeared following phylogenetic correction using pANCOVA), indicating that the distal limb of mammals possesses relatively greater fascicle area.

Pan-amniote regression

Coefficients for the pan-amniote regressions (Table 6) provide an empirical basis for estimating some important measures of muscle mass and force-generating capacity in extinct terrestrial amniotes. Note that the coefficients reported here were computed to the exclusion of bipeds, given that bipeds have been shown above to frequently differ from quadrupeds, and that the majority of species throughout tetrapod history were quadrupedal. The code provided in the Supplemental Information enables for predictive relationships to be derived that includes bipeds in the dataset. In addition to the coefficients, mean percent prediction error (%PE) is also reported, expressed in terms of the original, non-log-transformed dimensions of the data (Smith, 1980); this provides an alternative to prediction intervals (see SI code) as a way of deriving upper and lower estimates for a given taxon. Coefficients computed when the CFL was excluded from the reptile dataset (Table S10) are largely similar to those reported in Table 6, although notably mean %PE is generally higher.

Discussion

Through a synthesis of new dissection data with previously published results, the present study aimed to holistically appraise limb muscle scaling in terrestrial amniotes. In particular, it sought to investigate how muscle mass (size) and force-generating capacity (strength) scales at the level of the whole limb, to explore whether this reflects previously observed patterns noted for individual muscles or functional groups across disparate clades, and to assess how tightly constrained amniote limbs are in terms of whole-limb muscular composition. A subsidiary objective was to generate statistical relationships that have predictive value in inferring function in extinct amniotes. In synthesizing data from numerous studies, it must be acknowledged that the resulting dataset will likely contain a certain level of ‘noise’ due to various sources. One pertinent source of error is the measurement of fascicle length in fresh specimens: as was the case in the present study, this is typically undertaken following removal of a given muscle from the limb, whereupon the fascicles may non-systematically deviate from a ‘reasonable’ or functionally relevant length. This problem may be partially mitigated by using formalin-fixed specimens, where the limb joints can be locked in physiologically realistic poses prior to fixation (thus limiting fascicle length change after the muscle is removed), although this approach can involve its own set of challenges, such as muscle shrinkage (Kikuchi & Kuraoka, 2014; Martin et al., 2020). Digital methods that permit architectural measurement in situ, such as contrast-enhanced scanning and automated fascicle tracking (e.g., Sullivan et al., 2019), may be able to avoid these issues, but thus far are limited in spatial scale and hence anatomical and taxonomic scope. Additional sources of noise in the present study will also include variation in the approaches of prior studies, in terms of investigator, occasional exclusion of some small muscles, subjective subdivision of muscle complexes and so on. Nonetheless, given the wide diversity and size range of species covered here, this was considered acceptable for the present study’s broad scope.

Mass and force-generating capacity scaling across terrestrial amniotes

In terms of total muscle mass (Fig. 2), the present study found that reptile (i.e., non-avian saurian) limbs typically show negative allometry with respect to body mass (exponents 0.884 to 0.939), whereas mammals (exponents 0.989 to 1.019) and bipeds (exponents 1.104 to 1.109, but wide CIs) typically displayed isometry. This is in partial agreement with prior studies, where weak negative to modest positive allometry has been recovered (exponents 0.9 to 1.15 across all amniotes; see references in Introduction). The differences in findings may be due to multiple factors, including variation in sample sizes affecting CI calculation, the line-fitting approach used and even the species that contribute to the underlying datasets. It may also reflect a genuine biological phenomenon, where allometric patterns observed for a select few muscles (i.e., those which extend the range of exponents recovered by previous studies) are ‘cancelled out’ by those of many other muscles, when considered together at the level of the whole limb. Irrespective of the proximate cause(s), it is evident that reptiles have proportionately less muscle mass than mammals or bipeds, and bipeds have proportionately greater hindlimb muscle mass, especially in the distal limb. Neither result is hardly surprising, given that the reptiles sampled have short limbs and a long, massive tail that contributes substantially toward total body mass, and that bipeds (by virtue of being bipeds) ought to invest a greater fraction of body mass into longer, more heavily muscled hindlimbs.

Muscle force-generating capacity in the present study was expressed in a more intuitive fashion by normalizing to units of body weight. Thus, to compare exponents derived here to those of prior studies that have examined PCSA directly, this requires adding or subtracting 1.0 to the exponent. Three different measures of normalized force-generating capacity (Fmax*) were explored here—mean (Fig. 3), median (Fig. S2) and total (Fig. S3)—which showed a consistent overarching set of patterns with respect to body mass. Mammals and reptiles overall exhibit isometric scaling, although in some circumstances mammals tended towards positive allometry (total range of exponents −0.30 to −0.245) whereas reptiles tended towards negative allometry (total range of exponents −0.465 to −0.302). This is partially consistent with the results of prior studies (corrected exponents −0.31 to −0.09; see references in Introduction). Again, discrepancies in findings may belie the different spatial scales concerned; for example, the high end of previously reported exponents derive from the plantaris muscle of the mammalian hindlimb (Pollock & Shadwick, 1994) and the proximal forelimb of felids (Cuff et al., 2016a). Of note, reptiles often displayed a markedly lower intercept than mammals or bipeds, indicative of reduced relative force-generating capacity and again consistent with possessing short limbs and massive tails. It therefore follows that how ‘body mass’ is gauged (total body mass, body mass excluding tail, etc.) will influence interpretations of interspecific differences in force-generating capacity, although from a purely mechanical perspective total body mass is appropriate in the context of terrestrial locomotion, since ultimately the limbs must support and propel all of it.

Bipeds were found to almost ubiquitously display positive allometry in muscle force-generating capacity, and were frequently found to significantly differ from mammals and reptiles even after phylogeny was accounted for (in 13 of 15 comparisons with mammals, all comparisons with reptiles); again, this can be related to the greater investment of biomass in the hindlimbs of these species. Notably, positive allometry was exceedingly strong in the distal hindlimb (total range of exponents −0.016 to 0.025), to the point that relative force-generating ability may increase with increasing body size. This is comparable to exponents reported for bipedal hopping macropods (corrected exponents −0.22 to 0.33; Bennett & Taylor, 1995; McGowan, Skinner & Biewener, 2008), although intriguingly is higher than the exponents recovered by Maloiy et al. (1979) for the ankle extensors of a selection of ground-dwelling birds (corrected exponent −0.24 to −0.19). These results should be viewed with some caution, given that only six species were examined in the present study. Nonetheless, they raise the interesting question of whether extreme positive allometry would persist at larger body sizes beyond the largest extant biped, the ostrich (Struthio camelus). For example, how would muscle strength fare in multi-tonne non-avian theropod dinosaurs, and what implications may this have for locomotor performance? These are questions that must await future investigation, but the comparative dataset marshalled here can provide an empirical basis for such efforts.

In lieu of investigating fascicle length per se, this study examined an equivalent region-level metric, characteristic fascicle length (L*), which tended to exhibit negative allometry with respect to body mass in both mammals and bipeds, as well as the forelimb of reptiles. The exponents obtained (0.171 to 0.324) fell within the range reported previously for actual fascicle length in individual muscles or functional groups (0.14 to 0.5; see references in Introduction). Negative allometry indicates that muscle fascicles are on the whole proportionately shorter in larger species. This is especially true in the mammalian distal hindlimb, which also exhibits a lower intercept (Fig. 4); shorter muscle fascicles in the distal hindlimb of mammals in turn translate to relatively greater PCSA per unit mass in this compartment (Fig. 5). All else being equal, proportionately shorter muscle fascicles in larger species will translate to reduced joint excursion, and at least for mammals and birds this correlates with the habitual use of more extended limb postures in larger species (Biewener, 1989; Biewener, 2005; Bishop et al., 2018; Gatesy & Biewener, 1991).

Fascicle packing

Analysis of covariance of muscle mass and force-generating capacity identified numerous differences in scaling patterns between mammals, reptiles and bipeds, but when phylogeny was taken into account the majority of these became statistically non-significant (Tables 2, 3, Tables S2, S3). Hence there are genuine physical differences in mass or force-generating capacity between each group (which is biomechanically relevant), and these largely exist due to each group having evolved along different phylogenetic trajectories. However, when total muscle mass was factored into consideration, either directly (comparison #4: ΣPCSA v. Σmmuscle) or indirectly (comparison #3: L* v. mbody, wherein Σmmuscle is used to compute L*), ANCOVA recovered almost no statistically significant difference between any group; this was true regardless of whether phylogeny was accounted for or not, and was observed for the limb as a whole as well as proximal and distal compartments separately (Tables 4, 5). That is, the fore- and hindlimbs of reptiles, mammals and bipeds almost ubiquitously follow a single common scaling pattern1 . This in turn implies that observed differences in whole-limb force-generating capacity between groups are principally driven by differences in limb muscle mass, not internal architecture: one group does not ‘pack their fascicles better’ for a given volume of limb muscle, they just invest a greater fraction of total biomass into limb muscle in the first instance. The only possible exception to this generality is the distal hindlimb of mammals, which at larger body sizes at least do indeed appear to ‘pack their fascicles better’, with a greater PCSA (and in turn, force-generating capacity) per unit muscle mass, correlating with proportionately shorter fascicles for a given body size. All else being equal, this result suggests that larger species may benefit from greater elastic energy storage in the distal hindlimb tendons during locomotion (cf. Pollock & Shadwick, 1994), although it should nevertheless be treated with caution, since a large proportion of the currently sampled mammals are ‘cursorial’ taxa, which may bias the analysis.

The existence of a single overarching pattern, across a diverse array of terrestrial amniotes that span more than four orders of magnitude in body mass, is remarkable. It implies the tendency towards some adaptive optimum in organismal ‘design’ or, more likely, the presence of one or more constraints that prevent significant and systematic deviation from a common pattern. These constraints may be functional in nature, such as trade-offs that could occur between conflicting requirements of individual muscles in the execution of disparate tasks (see Introduction), or may have a developmental basis (e.g., Evans et al., 2021). Given the multidimensional and nonlinear aspects of muscle architecture and function, and terrestrial locomotor biomechanics in general, it would be naïve to suggest that a bivariate statistical model such as those derived here can sufficiently represent the mechanical phenomena involved (Taylor & Thomas, 2014). Deciphering the proximate underlying cause(s) of this strong consistency must therefore await future study. It is also important to recognize that there is still scope for (non-systematic) variation; for example, the greyhound (Canis familiaris, 27 kg) and snow leopard (Panthera uncia, 36 kg) have near-identical total hindlimb muscle mass (~1.64 kg), yet the greyhound—which is selectively bred for high-speed running—has approximately double the PCSA of the snow leopard (0.028 v. 0.014 m2; Table S1). Important insight into the fascicle packing problem can therefore be gained by better understanding the reasons for variation about the common pan-amniote pattern, through exploring variables that were not investigated in the current analyses, such as segment lengths, muscle moment arms, joint mobility, posture and locomotor ecology.

Considerations for future studies

A number of other points are worth noting in the context of future comparative investigations of muscle architecture. Firstly, despite a superficially broad taxonomic coverage, the dataset of the present study is still biased, with few birds, marsupials or non-varanid squamates, numerous ‘cursorial’ eutherians and not one testudine; future studies should therefore target currently under- or unsampled parts of amniote phylogeny. Increased diversity will be especially useful for elucidating the true extent, and underlying cause(s) of, conservatism in whole-limb fascicle packing. Furthermore, it is recommended that as much architectural data be collected and reported as possible, even if it may not all be immediately used in a given study, since this will maximize its potential utility for later studies. One reason why several prior datasets were excluded from the present study was their omission of pennation angle from the set of reported (or even collected) measurements. A second point worth considering is that, out of necessity, the present study ignored the intrinsic musculature of the manus and pes; if further datasets for these muscles become available in the future, they should be analysed, since this may reveal important differences between plantigrade and digitigrade species.

One final consideration is that the present study found numerous differences between mammals, reptiles and bipeds in the allometry of muscle mass and force-generating capacity with respect to body mass. This is hardly surprising, but it has an important implication for how comparative studies should undertake their analyses. In order to facilitate comparison within or across species, previous studies that have explored muscle architecture and function typically normalize raw architectural measurements by body mass, or the dimensionally appropriate exponent of body mass such as mbody⅓ for normalized fascicle length (Allen et al., 2010; Bates & Schachner, 2012; Dick & Clemente, 2016; Fahn-Lai, Biewener & Pierce, 2020; Martin et al., 2019; Regnault et al., 2020). This may be acceptable for comparisons of closely related species, but becomes questionable when divergent allometries with body mass are involved. A more sensible approach for future studies would be to normalize against more directly relevant parameters, such as limb or limb segment length, or alternatively by the appropriate clade-specific scaling exponents. Exactly which is the most appropriate normalizing metric to use may vary depending on a given study’s core questions, and deserves further scrutiny.

Conclusion

Whole-limb scaling of muscle mass and force-generating capacity generally follows the same patterns reported previously for individual muscles or functional groups of muscles, although some instances of tendency towards isometry (as opposed to previously reported positive or negative allometry) were noted. This is the first time that muscle scaling has been addressed at such a broad spatial scale, across a diverse array of terrestrial amniotes. Additionally, some important differences were observed between proximal and distal limb compartments, particularly in the hindlimb of mammals and bipeds, which may reflect the ‘cursorial’ habits of many of the species investigated here. The mammalian distal hindlimb has proportionately less muscle mass and shorter fascicles, but per unit muscle mass has a higher PCSA; and the distal hindlimb of bipeds has proportionately greater muscle mass and PCSA.

Almost all differences in force-generating capacity between groups appear to be due principally to differences in muscle mass, rather than muscle architecture. Thus, one group does not systematically ‘pack their fascicles better’ than another, instead they simply invest more biomass into limb muscle. The underlying reason(s) for a single overarching relationship across extant amniotes remains to be determined, although it echoes conservatism in other aspects of musculoskeletal design, such as tissue mechanical properties.

A comparative dataset of extant amniotes spanning almost five orders of magnitude in body size has been assembled, which can be built upon and used by future studies in the analysis of muscle architecture diversity. This dataset also forms the basis for a suite of pan-amniote predictive equations that can be used to estimate bulk muscle mass and force-generating capacity, along with error margins, for extinct species. These can be used if estimates of body mass are available, or alternatively from estimates of total limb muscle volume, as derived from digital volumetric reconstructions based on skeletal material. Reliably inferring the force-generating capacity of individual muscles in extinct species remains elusive (Bishop, Cuff & Hutchinson, 2021a), but inferences of total limb muscle force-generating capacity provides a step closer toward achieving this goal.

Supplemental Information

Supplemental Information 1 Supplemental figures and tables, and description of the supplemental code.

Click here for additional data file.

Supplemental Information 2 The raw architectural data used in this study.

Raw hindlimb data, raw forelimb data, hindlimb data collated into the parameters investigated in the study, and forelimb data collated into the parameters investigated in the study.

Click here for additional data file.

Supplemental Information 3 Supplemental code and data.

This contains R computer code used to perform all analyses in the study, along with necessary input data files (muscle data in .csv format, phylogenetic tree in .nwk format). The code also allows for estimation of a particular variable via the pan-amniote regression given some user-supplied input (e.g., body mass estimate for an extinct species), along with 95% prediction intervals.

Click here for additional data file.

T. French (Massachusetts Division of Fisheries and Wildlife), R. Reed (United States Geological Survey, Daniel Beard Center) and P. Fahn-Lai are thanked for providing the opossum and tegu specimens dissected here; P. Bergmann, C. Clemente, S. Cox, M. Martin and J. Rubenson are thanked for providing the original architectural data from their prior studies; and J. Hughes is thanked for logistical assistance with dissections. The constructive comments of two anonymous reviewers, and the editor, on a prior version of the manuscript are also greatly appreciated.

Additional Information and Declarations

Competing Interests

Author Contributions

Data Availability

1 This might at first seem at odds with the finding that, for example, mammals exhibit positive allometry whereas reptiles do not show a significant departure from isometry. The apparent contradiction can be reconciled by noting that these comparisons are testing two different things in isolation of one another, whereas in reality there is a gradation across successively overlapping exponents and CIs—the statistical analogue of a ‘ring species’.

Stephanie E. Pierce is an Academic Editor for PeerJ.

Peter J. Bishop conceived and designed the experiments, performed the experiments, analyzed the data, prepared figures and/or tables, authored or reviewed drafts of the paper, and approved the final draft.

Mark A. Wright performed the experiments, analyzed the data, authored or reviewed drafts of the paper, and approved the final draft.

Stephanie E. Pierce conceived and designed the experiments, authored or reviewed drafts of the paper, and approved the final draft.

The following information was supplied regarding data availability:

All data and R code used in the present study and additional figures and tables providing a further break-down of results reported in the article are available in the Supplemental Files.

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
