# Peer review of "Whole-limb scaling of muscle mass and force-generating capacity in amniotes"

_PeerJ, doi:10.7717/peerj.12574_

## Round 0.1 · original submission · Major Revisions

I received two detailed revisions of your work. I agree with our first reviewer that more clarity is required regarding the anatomical data; please follow the recommendations of this review. Our second reviewer pointed out another number of issues that need your further attention. I think that a better description of your methods is required. Following their suggestions will highly improve this interesting paper.

Reviewer 1 ·

Basic reporting

no comment

Experimental design

The statistical analyses presented here on the primary and secondary anatomical data are well thought out valid. However, I do believe that some clarification is required in regards to the anatomical data itself, both those collected for this study and those obtained from previous work. And while I don’t believe any assumptions made regarding the use of these data would have had any major bearing on the study results, I feel that the manuscript will benefit from these additional clarifications.

These include:

Primary anatomical data
- Of course, measuring a certain amount of fascicles in order to generate a mean value and represent a muscle fibre length is standard practise in many muscle architecture studies. However, as these were fresh specimens, was any consideration paid to the fact that once the muscles were removed from limb, their fibres may no longer be at an “anatomically reasonable” or functional relevant length? This is usually mitigated in some studies by formalin-fixing the limb with joint angles locked at certain positions, which limits the changes in fibre lengths when muscles are removed, although this potentially introduces other uncertainties due to muscle shrinkage. I’m aware that in the Discussion you mention the potential “noise” in this dataset due to the different methods used, among other things, but I do think that this potential limitation of the primary data you present here is worthy of specific mention, particularly if it is to be used in future studies.
- I also think it would be beneficial to report your primary anatomical data in the supplemental data as means ± standard deviations, as this would give an indication of the variation of fibre lengths, pennation angles and PCSAs within the species you investigated.

Bipeds
-Is there a specific justification for grouping humans with birds as “bipeds”, beyond their shared gaits? One would assume that as their limb postures are very different, their muscle properties, particularly in the proximal limb, would be similarly disparate. I’m not necessarily arguing that you remove humans completely, however I feel that a short sentence justifying this grouping could be beneficial, as it could explain your wide CIs

Validity of the findings

no comment

Additional comments

The manuscript is nicely written with a clear introduction to the problem of a lack of knowledge of how muscle force generating properties scale across species, which introduces issues when attempting to predict these factors in extinct species. The interpretations of the results are clear and effectively related to many uncertainties which persist regarding muscle architecture and function (i.e. fascicle packing, tendon energy storage). I have no major comments beyond some additional clarifications of certain aspects of the anaomical data.

Reviewer 2 ·

Basic reporting

no comment here

Experimental design

no comment here, more detail on the methods are needed as described in section 4

Validity of the findings

no comment, but further justification behind some findings could improve the manuscript, see section 4

Additional comments

The authors sought to review recent work on the scaling of the fore and hindlimb musculoskeletal system as well as add their own data from tegus and opossums to determine if there are general patterns to how muscle strength varied against predictions with body size in amniotes generally. This paper is overall a great contribution to the literature and I am glad to see this sort of comparative analysis being conducted.

Although the overall approach is sound, the authors should be careful to identify the assumptions in their analyses and the true meaning of the shorthand they employ. For example, the authors make frequent references to ‘strength’, by which they mean either physiological cross-sectional area (PCSA), or PCSA normalized to body weight. Although it is unwieldy, a more accurate term for this value would be anatomical force-generating capacity, or something similar?. Similarly, it is not necessary to refer to muscle mass as ‘size’, because mass is more precise (cannot be confused with volume) and is already simple enough to be easily understood by readers.

Second, the authors present here many valuable analyses but need to do a better job – both textually and graphically – at pointing the reader to the most salient findings. The high number of tables and graphs will serve as excellent reference material but should be presented in a way that better indicates what the most important findings are. Figures 3-5 especially are somewhat redundant. I think it would be best for the authors to determine which of the 3 figures sends the most important message and put the other two figures in the supplemental material.

Finally, the authors state that they are curious about “Are there different strategies for packing PCSA into a given volume of limb muscle, or do the differences observed between individual muscles and between species ‘cancel out’ at the level of the whole limb?” This is a good question, but their general conclusion on line 624 could use more support. Could the authors show how this trade off plays out within each limb segment, or does it only remain universal at the level of the whole limb? The general conclusion made in this section is a bold and important claim and should be properly justified.

Abstract

Methods

It is not fully clear here how averages were calculated. Were averages calculated for each muscle and then each species? Were raw values averaged and then exponents calculated? This information may seem pedantic but will be vital for replication studies. What about when multiple individuals were included for each species? How were the individuals treated differently? Also, were the original values back-calculated to raw if the cited publication presented mass-scaled values? Looking at the spreadsheet, it seems that the largest individual from literature sources was this. This choice should be explained and justified. It seems to me that this choice would a) increase the chance of erroneous measurements being included instead of being averaged out from multiple individuals, b) this could set up a false equivalence because, by aiming for the largest individual of each study, it may compare some large individuals with some smaller individuals from studies which do not feature large adults. In this case, it would probably be more advisable to average all individuals from a species to give each species an equal influence. Whatever course is taken, the methods must be completely clear to the reader because they will influence the results.

Also, please explicitly justify each of your isometric exponents through dimensional analysis.

line 177 – ignoring pes and manual muscle contributions would have a differential effect on plantigrade vs digitigrade species – you should acknowledge the impact of this in the discussion

line 290 – Great idea, but other differences between the groups should be mentioned (lighter skeletons in birds, weight of feathers in birds).

line 295 – the use of mean size-normalized isometric strength is a good idea but needs more defence/qualification. How would the meaning of this value change with differences in body posture and among muscles with different functional roles? I realize the muscles are being averaged, but this glosses over some important functional considerations.

line 298 – PCSA is actually a pretty good measure of area. Did you mean force here?

line 348 – Is isometry meant here to mean no change in pennation angle with muscle size? If so, why not measure that, or at least mention the similarity here.

Results

403 – re-define CFL here

410 – listing the non-mathematical version of each section heading also, e.g. “Total Muscle Mass vs. Body Mass” will substantially increase the impact of the study to non-specialists.

416 – what is meant by Cs ?

419 – starting with “reptiles have a lower…” is this for the phylogenetically-corrected analysis or not?

512 – Call these Quadruped Amniote regressions?

Discussion

lines 546 – can the CIs be included here as +/- values? This will help the readers to decide how much weight to put into your findings

lines 560 – can an estimate be made of how the weight of the tail affects these results? It would be very interesting if reptiles actually had less muscle mass than mammals in the limbs adjusted for tail weight, and by how much!

lines 564 – spell out body weight

619 – In this section especially it becomes unclear how results were calculated. Are these arguments being made for limb segments or for the limb as a whole, or for both limbs?

624 – That the relationship between PCSA and muscle mass does not vary between groups is an exciting finding. Could it mean that the statement on line 616 – that there are genuine physical differences in size or strength between each group – has more to do with how body mass is measured in birds or reptiles or mammals? Is it reasonable to say that reptiles are weaker than mammals because more muscle mass is put in the tail? For this reason I think the authors should 1) abandon the use of “strength” and “Size” as shorthand for their actual variables measured (PCSA, for example) and 2) further clarify and discuss how the study design affected the findings.

624 – This claim might be better supported by making comparisons of individual limb segments here in addition to whole-limb strength.

627 – it should be emphasized here that you are calculating relative biomass. Other factors that may influence

678 – If you are not sure what would be the most appropriate scaling base measurement, why not suggest that authors always report the raw data as well as the mass-scaled version?

There should be a table of the muscles included and classified as to whether they are proximal or distal that also defines the species codes in the supplemental excel file.

Figures / Tables
Figure 2 – For ease of comparison, please place the whole, proximal, distal labels on the left margin. Please edit the axis legends to be more readable such as log 10 Muscle Mass (kg) to facilitate quick comparison between different figures for the reader. Can the intercepts of the lines of isometry be moved to overlap with the plotted data? They would be much more useful this way! Can the same axis range be used for all plots? This will help in quick visual comparison of different slopes, which is the main point of these figures anyway. Can the font sizes of the slopes in the figure legends be increased and number of observations be listed as (n) ?

Figure 3 – Same comments as in Figure 2.

Table 8 – Can the slopes be listed before (to the left of) the intercept? They are more important and should be easier to read. Can the slopes of isometry also be listed here for easy comparison?

---

## Round 0.2 · Minor Revisions

I agree with both reviewers in that your study is an outstanding contribution to the field. There is still an issue that concerns me related to using reptiles instead of non-avian reptiles. You use reptiles as a common name, not the clade name Reptilia and this is no problem. However, this can cause misunderstandings if people are not paying attention to your explanation. Please consider this point before keeping the word reptiles.

The epigraph of the cladogram of figure S1 does not apply to the legends on the right. Biped is not a taxon name. You should write Aves. The same is valid for H. sapiens. It seems as if humans were not mammals. Please, try to find a way to be consistent. Thank you.

Reviewer 1 ·

Basic reporting

No comment

Experimental design

No comment

Validity of the findings

No comment

Additional comments

No comment

Reviewer 2 ·

Basic reporting

All correct.

Experimental design

All good here, my comments have been addressed.

Validity of the findings

All good here, my comments have been addressed.

Additional comments

Thanks to the authors for addressing my concerns and questions. They may wish to cite a new paper (not my own) ond different methods used to calculate PCSA in their relevant section ~ line 220.

Martin ML, Travouillon KJ, Fleming PA, Warburton NM. 2020 Review of the methods used for calculating physiological cross-sectional area (PCSA) for ecological questions. J. Morphol. 281, 778–789. (doi:10.1002/jmor.21139)

---

## Round 0.3 · accepted · Accept

Thank you for your patient. You find a clever solution to the point I highlighted. We can now go ahead.